# Butyrate Prevents TGF-β1-Induced Alveolar Myofibroblast Differentiation and Modulates Energy Metabolism

**DOI:** 10.3390/metabo11050258

**Published:** 2021-04-22

**Authors:** Hyo Yeong Lee, Somi Nam, Mi Jeong Kim, Su Jung Kim, Sung Hoon Back, Hyun Ju Yoo

**Affiliations:** 1Department of Convergence Medicine, Asan Institute for Life Sciences, Asan Medical Center, University of Ulsan College of Medicine, Seoul 05505, Korea; lhy_1279@naver.com (H.Y.L.); su-jungea@hanmail.net (S.J.K.); 2School of Biological Sciences, University of Ulsan, Ulsan 44610, Korea; somisomi1230@gmail.com (S.N.); wlsaud8787@naver.com (M.J.K.)

**Keywords:** short chain fatty acids, butyrate, TGF-β1, pulmonary fibrosis, myofibroblast differentiation, mitochondria, metabolites

## Abstract

Idiopathic pulmonary fibrosis (IPF) is a serious lung disease characterized by excessive collagen matrix deposition and extracellular remodeling. Signaling pathways mediated by fibrotic cytokine transforming growth factor β1 (TGF-β1) make important contributions to pulmonary fibrosis, but it remains unclear how TGF-β1 alters metabolism and modulates the activation and differentiation of pulmonary fibroblasts. We found that TGF-β1 lowers NADH and NADH/NAD levels, possibly due to changes in the TCA cycle, resulting in reductions in the ATP level and oxidative phosphorylation in pulmonary fibroblasts. In addition, we showed that butyrate (C4), a short chain fatty acid (SCFA), exhibits potent antifibrotic activity by inhibiting expression of fibrosis markers. Butyrate treatment inhibited mitochondrial elongation in TGF-β1-treated lung fibroblasts and increased the mitochondrial membrane potential (MMP). Consistent with the mitochondrial observations, butyrate significantly increased ADP, ATP, NADH, and NADH/NAD levels in TGF-β1-treated pulmonary fibroblasts. Collectively, our findings indicate that TGF-β1 induces changes in mitochondrial dynamics and energy metabolism during myofibroblast differentiation, and that these changes can be modulated by butyrate, which enhances mitochondrial function.

## 1. Introduction

Pulmonary fibrosis develops when lung fibroblasts undergo abnormal activation, proliferation, and differentiation toward myofibroblasts, resulting in excessive accumulation of extracellular matrix proteins, including collagen and tissue remodeling [1]. The damaged and stiff lung tissues interfere with proper organ function. Pulmonary fibrosis could be caused by infection, inflammation, environmental pollutants, and radiation exposure; however, in most cases, the causes are not well understood [2]. Myofibroblast differentiation and activation by transforming growth factor-β1 (TGF-β1) is a critical event in the pathogenesis of human lung fibrosis [3,4,5,6]. TGF-β1 induces the expression of cytoskeletal proteins such as α-smooth muscle actin (α-SMA) and extracellular matrix proteins, such as type I collagens and fibronectin [7,8].

Previously, the respiratory system was thought to be sterile, but many recent studies have confirmed the existence of a microbiome in the respiratory tract; bacterial 16s-rRNA genes have been used to identify the constituent species [9]. Respiratory microbiota could be involved in maintaining homeostasis of respiratory physiology, and this affects the pathogenesis of respiratory disease [10]. Higher bacterial burden is present in bronchoalveolar lavage fluid from patients with idiopathic pulmonary fibrosis (IPF), and the diversity and composition of microbiota are related to the levels of alveolar profibrotic cytokines [11,12,13]. Cohort studies have demonstrated that disease progression in patients with IPF was associated with lung microbiome [13,14,15]. Bacteria have the potential to cause cell injury in the airways, and host–microbiome interactions might influence fibroblast responsiveness and cytokine expression by bacterial metabolites, mainly short chain fatty acids (SCFAs) [10,16]. In addition, a recent study reported that SCFAs produced by lung anaerobic bacteria in HIV patients could affect immune response in infectious lung disease [17].

SCFAs, which are primarily produced by bacterial fermentation of dietary fibers in the human body [18], are taken up by the host and used as energy sources or regulators [19]. The main SCFAs are acetate (C2), propionate (C3), and butyrate (C4), together constituting 95% of total SCFAs. SCFAs are metabolized at various sites in the body, transported from the intestinal lumen into the blood, and present in various tissues [19]. SCFAs can be found at tens to hundreds mmole/L concentrations in human intestine, and at several mmole/L concentrations in human serum and pulmonary airways [20,21,22,23]. Mice intestine and lung have been known to contain approximately 10 μmoles of SCFAs per gram of tissue [24,25]. SCFAs play important roles in regulating host energy metabolism, insulin sensitivity, and immune responses [26,27,28]. Circulating levels of SCFAs could shape the immunological environment in the lung and influence the severity of airway allergic inflammation [16].SCFAs can be present in lung tissues, either due to the presence of SCFAs in the bloodstream and/or generated as metabolic products by microbiota residing in the respiratory system. Thus, investigation of the effect of SCFAs on pulmonary fibrosis would aid in understanding the roles of the microbiota and their metabolic products in lung fibrosis [29]. Mitochondrial dysfunction may be involved in the pathogenesis of pulmonary fibrosis [30,31,32]. Mitochondrial damage, depletion of ATP, and activated glycolysis have been observed in fibrotic lung [33,34]. In this study, we investigated the effect of SCFAs on changes in fibrotic marker expression and metabolites involved in energy metabolism during TGF-β1-induced myofibroblast differentiation.

## 2. Results and Discussion

### 2.1. TGF-β1 Treatment Increases Expression of Fibrotic Markers and Decreases Energy Metabolism in Pulmonary Fibroblasts

TGF-β1 plays a crucial role in the pathogenesis of pulmonary fibrosis, and its levels are elevated both in experimental models of pulmonary fibrosis and in human lungs. [8,35,36]. We treated MRC5 human pulmonary fibroblasts with three different concentrations of TGF-β1 (1, 2, and 5 ng/mL) and measured the expression levels of two fibrotic markers, Col1A1 and α-SMA, at several time points. The levels of fibrotic markers were upregulated in a dose- and time-dependent manner of TGF-β1 as expected (Figure 1A,B), although they were not strongly affected by different amounts of TGF-β1. Next, mRNA expression levels of multiple fibrotic marker genes (Col1A1, Col3A1, α-SMA, and Fibronectin) were measured in MRC5 cells treated with 1 or 5 ng/mL TGF-β1 at several time points (Figure 1C). As expected, TGF-β1 treatments strongly induced expression of all fibrotic marker genes in time-dependent manner, but mRNA levels of the fibrotic genes were not affected by different TGF-β1 concentrations at all time points (Figure 1C). Furthermore, immunofluorescence analysis of Col1A1 and α-SMA showed that TGF-β1 (5 ng/mL) treatment robustly induced intracellular accumulation of the fibrotic marker proteins compared to mock treatment in MRC5 cells, indicating that the concentration (5 ng/mL) of TGF-β1 was strong enough to induce differentiation of pulmonary fibroblasts into myofibroblasts (Figure 1D). Therefore, subsequent studies were conducted using 5 ng/mL TGF-β1.

Next, we performed targeted metabolomics to explore the changes in metabolites related to energy metabolism in TGF-β1-treated pulmonary fibroblasts. As shown in Figure 1E, TGF-β1-treated pulmonary fibroblasts produced less energy than mock-treated fibroblasts. ATP and NADH levels were significantly lower in TGF-β1-treated fibroblasts, and the difference in NADH levels between mock-treated and TGF-β1-treated pulmonary fibroblasts was increased over time. The measurement of NADH and NADH/NAD levels can be used to monitor the levels of oxidative phosphorylation. Thus, the lower NADH and NADH/NAD levels in TGF-β1-treated pulmonary fibroblasts represent reduced oxidative phosphorylation, implying mitochondrial dysfunction. Significant accumulation of dysfunctional mitochondria in fibrotic lungs is associated with the development of pulmonary fibrosis [30,34,37]. In addition, glycolysis is activated in in vitro differentiated myofibroblasts from patients with IPF and human fibrotic lung tissues [37,38,39]. The metabolites involved in energy metabolism that we examined are listed in Appendix A. Citrate, malate, and fumarate levels were reduced in TGF-β1-treated pulmonary fibroblasts, suggesting that the TCA cycle was suppressed; however, glycolysis was not activated. Rather, lactate levels were decreased by TGF-β1 treatment, although the reduction was not as significant as the reduction in metabolic intermediates in the TCA cycle. In biological systems, many reactions take place continuously, and individual metabolites are often involved in several metabolic pathways. Thus, metabolites can change rapidly, and their levels are considered to be highly dynamic. Taking a snapshot at one moment may not reflect the dynamic changes in metabolites. To better understand dynamic metabolic changes, it is necessary to perform metabolic flux analysis. Nonetheless, the fact that glycolytic intermediates were unchanged, whereas the levels of TCA cycle intermediates changed markedly, indicates that TGF-β1 would affect the TCA cycle to a greater extent than glycolysis in pulmonary fibroblasts.

### 2.2. Effect of SCFAs on Fibrotic Marker Expression of TGF-β1-Treated Pulmonary Fibroblasts

To evaluate the effect of SCFAs on activation and differentiation of TGF-β1-treated pulmonary fibroblasts, we applied three different SCFAs (C2 [sodium acetate], C3 [sodium propionate], and C4 [sodium butyrate]) to PBS- or TGF-β1-treated human pulmonary fibroblasts for 12 h, and then measured the mRNA expression of fibrotic markers Col1Al and Col3A1 (Figure 2A). The concentrations of SCFAs used in this study were chosen based on the concentrations found in human or mouse lung and pulmonary airways [21,22,24]. Accordingly, other research groups have performed cellular experiments with ones to tens of millimolar concentrations of SCFAs to explore their biological roles [40,41]. Both Col1Al and Col3A1 mRNA levels increased upon TGF-β1 treatment. Acetate (C2) had no effect on either mRNA level, irrespective of the presence of TGF-β1. On the other hand, butyrate (C4) significantly decreased both mRNA levels in both PBS- and TGF-β1-treated pulmonary fibroblasts. Furthermore, butyrate decreased the increase in Col1Al and Col3A1 mRNA levels in TGF-β1-treated vs. PBS-treated cells. Next, to further explore the effect of SCFAs on pulmonary fibroblasts, we measured the levels of fibrotic markers at different concentrations (1, 5, and 10 mM) of SCFAs and a longer treatment time (48 h) (Figure 2B). Acetate (C2) barely had an effect on the expression of fibrotic markers and propionate (C3) could not reduce fibronection expression in TGF-β1-treated pulmonary fibroblasts. However, butyrate (C4) decreased the mRNA levels of Col1A1, Col3A1, α-SMA, and fibronectin in a dose-dependent manner (Figure 2B). The effect of butyrate was maintained for 48 h and was stronger than those of other SCFAs (Figure 2A,B). The results shown in Figure 2 confirmed the antifibrotic effect of 5 mM butyrate (C4) in TGF-β1-treated pulmonary fibroblasts. Although TGF-β1 treatment of pulmonary fibroblasts gradually increased expression of fibrotic marker genes over time, butyrate dramatically inhibited the mRNA and protein levels of these fibrotic markers almost to basal levels, except in the case of fibronectin (Figure 3A–C).

### 2.3. Butyrate Enhances Mitochondrial Function in Pulmonary Fibroblasts Whose Differentiation Is Surpressed

Because TGF-β1 significantly inhibited the TCA cycle (Appendix A) and oxidative phosphorylation (Figure 1E) in pulmonary fibroblasts, whereas butyrate (C4) drastically suppressed myofibroblast differentiation, we analyzed the changes in mitochondria morphology and membrane potential in cells treated with TGF-β1 or the combination of TGF-β1 and C4. Interestingly, 24-h treatment with TGF-β1 increased the population of cells containing elongated mitochondria (Figure 4A). This increase in elongated mitochondria persisted through the 48-h treatment, indicating that this elongation reflected a stable remodeling of mitochondrial morphology. However, alteration of mitochondrial morphology was not observed in cells co-treated with butyrate (C4) and TGF-β1 (Figure 4A). Cells adapt to certain types of stress, including nutrient starvation, endoplasmic reticulum (ER) stress, UV light irradiation, transcription inhibition, and ribosome inhibition, by inducing protective mitochondrial elongation through mechanisms such as stress-induced mitochondrial hyperfusion (SIMH) [42,43,44,45]. TGF-β1-mediated mitochondrial elongation is also observed in cell types such as TGF-β1-induced senescent Mv1Lu lung epithelial cells [46] and retinal pigment epithelial ARPE-19 cells [47]. Therefore, it is possible that myofibroblast differentiation from pulmonary fibroblasts following TGF-β1 treatment could induce mitochondrial hyperfusion via a stress-adaptive pathway, whereas mitochondrial hyperfusion may be inhibited, or at least not required, in cells treated with butyrate (C4) because this compound suppresses TGF-β1-mediated myofibroblast differentiation. Because changes in morphology influence mitochondrial function [42,43,44,45], we next checked the mitochondrial membrane potential (MMP), a key parameter of mitochondrial function that serves as an indicator of cell health [48]. Following treatment with TGF-β1 or TGF-β1 + C4, changes in MMP levels were determined by fluorescence-based measurement using JC-1 dye, a lipophilic, cationic dye [49]. In energized mitochondria, the accumulated dye can be visualized as red fluorescent aggregates, whereas in cells with low MMP, it remains as the monomeric form, emitting green fluorescence. Therefore, mitochondrial depolarization is indicated by a decrease in the red/green fluorescence intensity ratio. The cells that were exposed to TGF-β1 + C4 for 24 h exhibited a significantly higher JC-1 Red/Green ratio (Figure 4B). The ratio was maintained for 48 h, although TGF-β1 treatment also increased the ratio at 48 h, indicating that butyrate (C4) treatment can increase MMP and keep the fibroblast healthy. To substantiate these observations, we measured changes in metabolites involved in energy metabolism in TGF-β1-treated pulmonary fibroblasts with or without butyrate (Figure 5 and Appendix A). From early (6 h) until late in treatment (24 h), butyrate significantly increased ADP, ATP, NADH, and NADH/NAD levels in TGF-β1-treated pulmonary fibroblasts. Most of the metabolites’ changes were not meaningful, but citrate and AKG were significantly increased upon butyrate (C4) treatment (Appendix A), indicating that butyrate (C4) boosts mitochondrial energy metabolism in pulmonary fibroblasts. Glycolysis and the TCA cycle are often coupled to other metabolic pathways, such as the synthesis of amino acids, and their metabolic intermediates might change dynamically, comparing to the metabolic pathways where interactions with neighboring metabolic pathways unlikely happen [50]. On the other hand, it was reported that fluctuation of water molecules or ATP (likely NADH) is negligible because these molecules are often used as side-reactants [50]. Thus, the changes of ATP and NADH were emphasized to draw a conclusion in our study. Together, these observations suggest that butyrate (C4) plays important roles in the regulation of mitochondrial dynamics and modulation of energy metabolism, which may relate to inhibition of TGF-β1-mediated myofibroblast differentiation. Interestingly, NADH/NAD levels in TGF-β1 + C2 and TGF-β1 + C3 were slightly higher than that of TGF-β1 only, although the change of NADH/NAD in TGF-β1 + C3 was not significant (Appendix A). SCFAs can be used as substrates for TCA cycle. Butyrate can feed the TCA cycle via acetyl-CoA after β-oxidation. Acetate and propionate can be converted to acetyl-CoA and succinyl-CoA, respectively [51]. Thus, the slight increase in NADH/NAD levels in TGF-β1 + C2 and TGF-β1 + C3 might be attributed by the increased acetyl-CoA and succinyl-CoA. Accordingly, acetyl-CoA converted by butyrate might contribute to the increased NADH/NAD in TGF-β1 + C4; however, the actual increase in NADH/NAD in TGF-β1 + C4 was much higher than those observed with C2 or C3. As shown in Figure 2, acetate (C2) barely had an effect on the expression of fibrotic markers, and propionate (C3) could not reduce fibronection expression in TGF-β1-treated pulmonary fibroblasts (Figure 2). However, butyrate (C4) reduced the expression of all fibrotic markers that we tested. Thus, we assumed that the increased acetyl-CoA supplied by C4 would not be a main cause for the modulation of energy metabolism by C4 in TGF-β1-treated differentiation. Instead, an alternative mechanism mediated by butyrate might inhibit TGF-β1-induced differentiation of pulmonary fibroblasts to myofibroblasts, which would lead to the restoration of mitochondrial function. Further studies would be required to explore the inhibitory mechanism of butyrate in TGF-β1-treated differentiation of pulmonary fibroblasts to myofibroblast.

## 3. Materials and Methods

### 3.1. Chemicals and Reagents

TGF-β1 was purchased from PeproTech (Rocky Hill, NJ, USA). The following were purchased from Sigma-Aldrich (St. Louis, MO, USA): C2 (sodium acetate), C3 (sodium propionate), and C4 (sodium butyrate), Hoechst 33258, 4′,6-diamidino-2-phenylindole (DAPI). MitoTracker Red and JC-1 were purchased from Molecular Probes (Eugene, OR, USA). Standard metabolites and internal standard were purchased from Sigma-Aldrich (USA) or CDN Isotopes (Pointe-Claire, Quebec, Canada). All stock solutions for liquid chromatography-tandem mass spectrometry (LC-MS/MS) were prepared in water/acetonitrile (50/50, *v*/*v*) and stored at −20 °C. All solvents including water were from J. T. Bakers (USA).

### 3.2. Cell Culture and Treatments

MRC5 human fetal lung fibroblasts (CCL-171, ATCC, Manassas, VA, USA) were cultured in MEM (Gyeongsan-si, Gyeongsangbuk-do, WelGENE, Gyeongsan, Korea) containing 10% fetal bovine serum (WelGENE, Gyeongsan, Korea) and 1% penicillin–streptomycin (WelGENE, Gyeongsan, Korea) at 37 °C in a 5% CO_2_ humidified environment. For myofibroblast differentiation, MRC5 cells were treated with TGF-β1 (1, 2, or 5 ng/mL) for the indicated times. To evaluate the anti-fibrogenic effects of SCFAs, cells were treated with TGF-β1 for indicated times with or without C2, C3, or C4.

### 3.3. Western Blot Analysis and Antibodies

Cells were lysed with NP-40 lysis buffer (1% Nonidet P-40, 0.05% sodium dodecyl sulfate, 50 mM Tris-Cl [pH 7.5], 150 mM NaCl, 0.5 mM sodium vanadate, 100 mM sodium fluoride, 50 mM β-glycerophosphate) supplemented with Halt Protease Inhibitor Cocktail (Thermo Fisher Scientific, Waltham, MA, USA). Homogenates were centrifuged at 12,000× *g* for 15 min at 4 °C, and supernatants were collected. The protein concentration was determined using a BCA protein assay kit (Thermo Fisher Scientific, Waltham, MA, USA). Protein samples were resolved by sodium dodecyl sulfate-polyacrylamide gel electrophoresis (SDS-PAGE) and transferred to polyvinylidene difluoride or nitrocellulose membranes. The membranes were blocked for 1 h at room temperature with 5% skim milk in Tris buffered saline-Tween buffer (0.1% Tween 20, 20 mM Tris-HCl, pH 7.5, 150 mM NaCl). Membranes were incubated with indicated primary antibodies (anti-β-actin [DSHB, Iowa city, IA, USA], anti-Col1A1 [SouthernBiotech, Birmingham, AL, USA], anti-α-SMA [Abcam, Cambridge, CB2 0AX, UK]) at 4 °C overnight and then with horseradish peroxidase-conjugated secondary antibody. The chemiluminescence signal was detected with an Azure Biosystems C300 Analyzer (Azure Biosystems, Dublin, CA, USA) using SuperSignal West Pico Chemiluminescent Substrate (Thermo Fisher Scientific, Waltham, MA, USA).

### 3.4. RNA Isolation and Quantitative Polymerase Chain Reaction (qPCR)

Total RNA was prepared from MRC5 cells treated with the indicated chemicals using the QIAzol Lysis reagent (QIAGEN, Hilden, Germany). cDNAs were prepared with a High-Capacity cDNA RT kit (Applied Biosystems, Foster City, CA, USA) for quantitative polymerase chain reaction (qPCR). The qPCRs were carried out using the Luna Universal qPCR Master Mix (NEB, Ipswich, MA, USA) with the appropriate primers on a StepOnePlus Real Time System (Applied Biosystems, Foster City, CA, USA). The specificity of each primer pair was confirmed using melting curve analysis. Copy number relative to β-actin mRNA was calculated as previously described [52] using the 2 method. Primer sequences were as follows: 5′-CCTGGATGCCATCAAAGTCT-3′ and 5′-CGCCATACTCGAACTGGAAT-3′ (Col1A1); 5′-CTTCTCTCCAGCCGAGCTTC-3′ and 5′-GTAGTCTCACAGCCTTGCGT-3′ (Col3A1); 5′-AATGCAGAAGGAGATCACGG-3′ and 5′-TCCTGTTTGCTGATCCACATC-3′ (α-SMA1); 5′-TGTCAGTCAAAGCAAGCCCG-3′ and 5′-TTAGGACGCTCATAAGTG-3′ (Fibronectin).

### 3.5. Microscopy and Image Analysis

For immunofluorescence, MRC5 cells were plated on collagen-coated glass cover slips in 60 mm dishes and cultured overnight. Two days later, the cells were treated with the indicated chemicals, fixed with 3.5% paraformaldehyde in PBS for 15 min, and permeabilized with 0.2% Triton X-100 in PBS for 2 min. The cells were blocked with 3% bovine serum albumin in PBS for 60 min and incubated with the indicated antibodies (anti-Col1A1 [SouthernBiotech, Birmingham, AL, USA] or anti-α-SMA [Abcam, Waltham, MA, USA]) overnight at 4 °C. The cells were further incubated with fluorescein isothiocyanate-conjugated secondary antibody at room temperature for 1 h. Nuclei were stained with DAPI. Finally, the cells were observed by confocal laser microscopy using a FV1200-OSR microscope (Olympus, Nishi-Shinjuku, Shinjuku-ku, Japan).

For mitochondrial observation, MRC5 cells were plated on a 35 mm cell imaging dish with a glass bottom coated with 0.01% collagen in PBS and cultured overnight. Two days later, the cells were treated with the indicated chemicals for the indicated times. After treatment, the cells were stained with Mitotracker Red (25 nM) for 20 min or JC-1 (1.38 μg/mL) plus Hoechst 33,258 (5 μg/mL) for 60 min in culture media, respectively. Fluorescence images from living cells were obtained using a FV1200-OSR confocal laser microscope. To quantify elongated, tubular, or fragmented mitochondria in Mitotracker Red-stained cells, we collected images for more than 30 cells per condition and scored the cells for the presence of primarily elongated, tubular, or fragmented mitochondria. Scores are presented in the bar graphs.

### 3.6. Liquid Chromatography-Tandem Mass Spectrometry (LC-MS/MS)

Cells were harvested using 1.4 mL cold methanol/H_2_O (80/20, *v*/*v*), after rapid sequential washing with PBS and H_2_O. The cells were lysed by vigorous vortexing, and 50 μL internal standard (10 μM of ^13^C_5_-glutamine) was added. Metabolites were extracted from the aqueous phase by liquid–liquid extraction after addition of chloroform. The aqueous phase was dried by vacuum centrifugation, and the sample was reconstituted with 50 μL of 50% methanol prior to LC-MS/MS analysis.

Metabolites were analyzed with LC-MS/MS equipped with 1290 HPLC (Agilent, Santa Clara, CA, USA), Qtrap 5500 (ABSciex, Framingham, MA, USA), and a LC column (Synergi fusion RP 50 × 2 mm, Phenomenex Inc., Torrance, CA, USA). Three microliter of sample was injected into the LC-MS/MS system and ionized with a turbo spray ionization source. Ammonium acetate (5 mM) in H_2_O and methanol were used as mobile phases A and B, respectively. The separation gradient was as follows: hold at 0% B for 5 min, 0% to 90% B for 2 min, hold at 90% for 8 min, 90% to 0% B for 1 min, and hold at 0% B for 9 min. LC flow was 70 μL/min for the entire run except between 7–15 min, when it was 140 μL/min, and column temperature was kept at 23 °C. Multiple reaction monitoring (MRM) was used in negative ion mode, and the extracted ion chromatogram (EIC) corresponding to the specific transition for each metabolite was used for quantitation. The area under the curve of each EIC was normalized against that of the EIC of the internal standard. The peak area ratio of each metabolite to the internal standard was normalized against the total amount of protein in the sample, and then used for relative comparison. Data analysis was performed using Analyst 1.5.2 software.

### 3.7. Statistical Analysis

All data are represented as means ± SEM of three or four independent experiments. The data were analyzed using GraphPad Prism 5 (GraphPad Software, Inc., La Jolla, CA, USA). Unpaired two-tailed Student’s *t*-tests were performed to determine the statistical significance of paired samples. *p* < 0.05 was considered significant.

## 4. Conclusions

Here, our results show that TGF-β1 decreases the NADH level in pulmonary fibroblasts, leading to lower rates of oxidative phosphorylation. Stress-induced mitochondrial hyperfusion (SIMH) may compensate for TGF-β1-mediated mitochondrial damage in pulmonary fibroblasts. However, butyrate (C4) blocks gene expression associated with TGF-β1-induced fibroblast activation. Furthermore, butyrate (C4) increases the MMP, boosts energy metabolic pathways that generate NADH and ATP, and keeps mitochondria healthy in TGF-β1-treated pulmonary fibroblasts. Together, our findings provide insight into the pivotal role of a specific metabolite from microbiota in the management of pulmonary fibrosis. Further studies are needed to explore the molecular mechanisms of the molecule for its potential clinical application.

## Figures and Tables

**Figure 1 metabolites-11-00258-f001:**
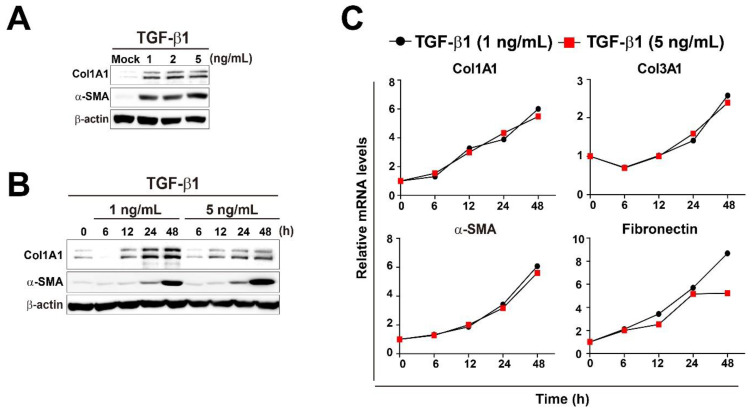
Changes in energy metabolites during TGF-β1-induced myofibroblast differentiation in MRC5 lung fibroblasts. (**A**) MRC5 cells were treated with TGF-β1 (1, 2, or 5 ng/mL) for 48 h. (**B**) MRC5 cells were treated with TGF-β1 (1 or 5 ng/mL) for the indicated times. Total cellular extracts were prepared and subjected to western blot analysis. (**C**) MRC5 cells were treated with TGF-β1 (1 or 5 ng/mL) for the indicated times. Total RNA was prepared and subjected to reverse transcription (RT) and qPCR analysis. (**D**) Expression of Col1A1 and α-SMA was observed by immunofluorescence in MRC5 cells that were treated with PBS (mock) or 5 ng/mL TGF-β1 for 48 h. Representative images are shown. (**E**) Measurement of ATP, ADP, AMP, ATP/ADP, ATP/(ADP + AMP), NAD, NADH, and NADH/NAD in mock- and TGF-β1-treated pulmonary fibroblasts. MRC5 cells were treated with PBS (mock) or 5 ng/mL TGF-β1 for the indicated times. The data are expressed as means ± SEM of three independent experiments. * *p* < 0.05 and ** *p* < 0.01; Mock-treated cells vs. TGF-β1-treated cells.

**Figure 2 metabolites-11-00258-f002:**
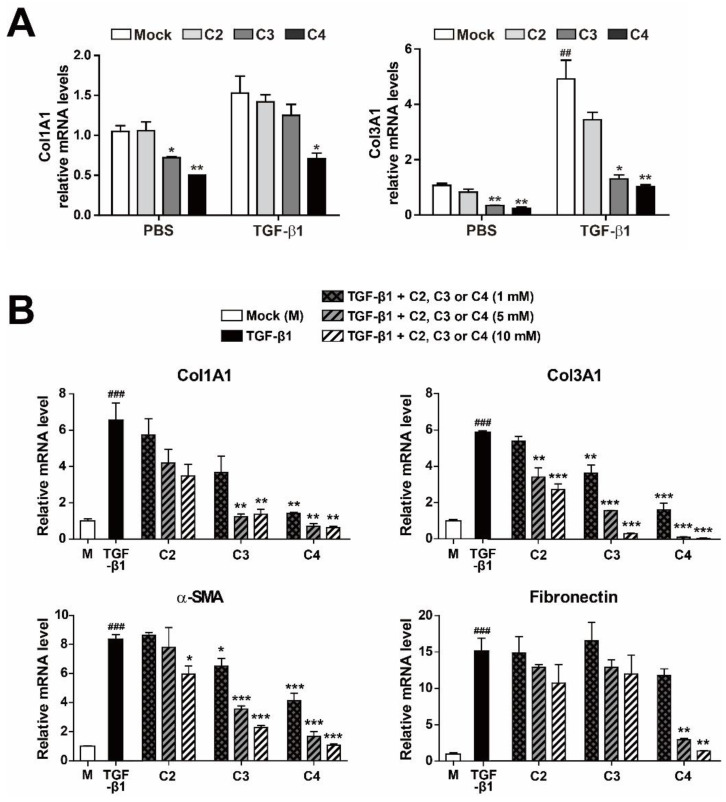
Inhibition of fibrotic marker expression by SCFA treatment of activated pulmonary fibroblasts. (**A**) MRC5 cells were treated with PBS or TGF-β1 (5 ng/mL) for 12 h in the presence of C2, C3, or C4 SCFAs (10 mM). Data are expressed as means ± SEM of three independent experiments, ^##^
*p* < 0.01; mock-treated cells from PBS-treated group vs. mock-treated cells from TGF-β1-treated group, * *p* < 0.05 and ** *p* < 0.01; mock-treated cells vs. C3 or C4-treated cells from each PBS or TGF-β1-treated group. (**B**) MRC5 cells were treated with TGF-β1 (5 ng/mL) for 48 h with the indicated amounts of C2, C3, or C4. Total RNA was prepared and subjected to RT and qPCR analysis for expression of fibrotic markers. Data are expressed as means ± SEM of three independent experiments, ^###^
*p* < 0.001; mock-treated cells vs. TGF-β1-treated cells, * *p* < 0.05, ** *p* < 0.01 and *** *p* < 0.001; TGF-β1-treated cells vs. TGF-β1 + C2, C3 or C4-treated cells.

**Figure 3 metabolites-11-00258-f003:**
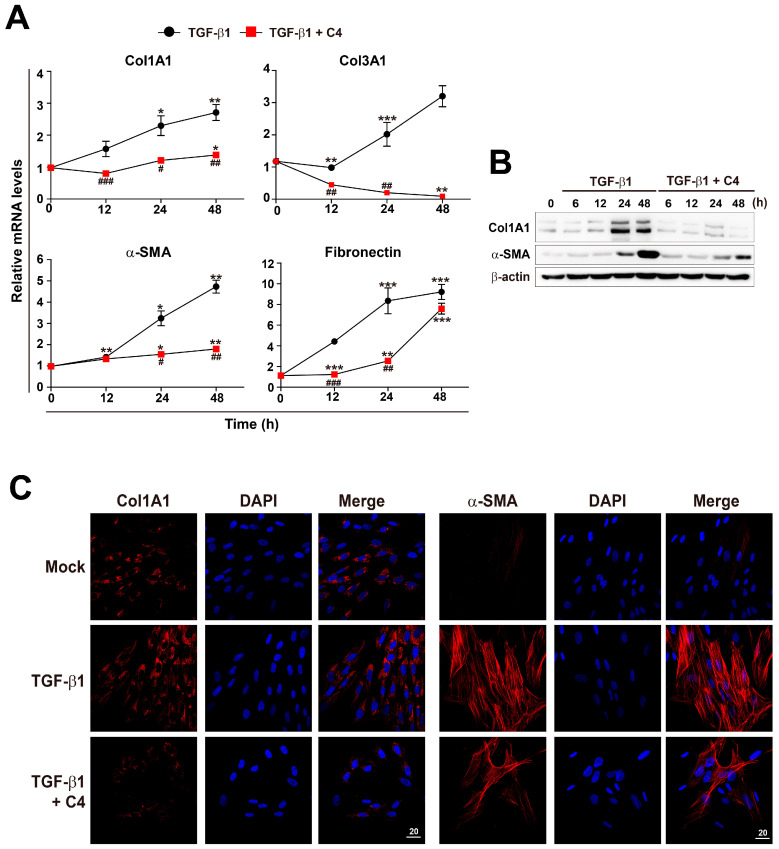
Inhibition of TGF-β1-mediated fibroblast activation by C4 treatment. (**A**) MRC5 cells were treated with TGF-β1 (5 ng/mL) for the indicated times with or without C4 (5 mM). (**A**) Total RNA was prepared and subjected to RT and qPCR analysis for expression of fibrotic markers. Data are expressed as means ± SEM of three independent experiments, * *p* < 0.05, ** *p* < 0.01, and *** *p* < 0.001; 0 h vs. each time point, ^#^
*p* < 0.01, ^##^
*p* < 0.05, and ^###^
*p* < 0.001; TGF-β1-treated cells vs. TGF-β1 + C4-treated cells. (**B**) Cells were treated with TGF-β1 (5 ng/mL) for the indicated times with or without C4 (5 mM), and total cellular extracts were prepared and subjected to western blot analysis. (**C**) Cells were treated with TGF-β1 (5 ng/mL) for 48 h with or without C4 (5 mM), and then stained with Hoechst 33,258 and anti-Col1A1 or anti-α-SMA antibody and analyzed on a confocal fluorescence microscope. Representative images are shown. Scale bar denotes 20 μm.

**Figure 4 metabolites-11-00258-f004:**
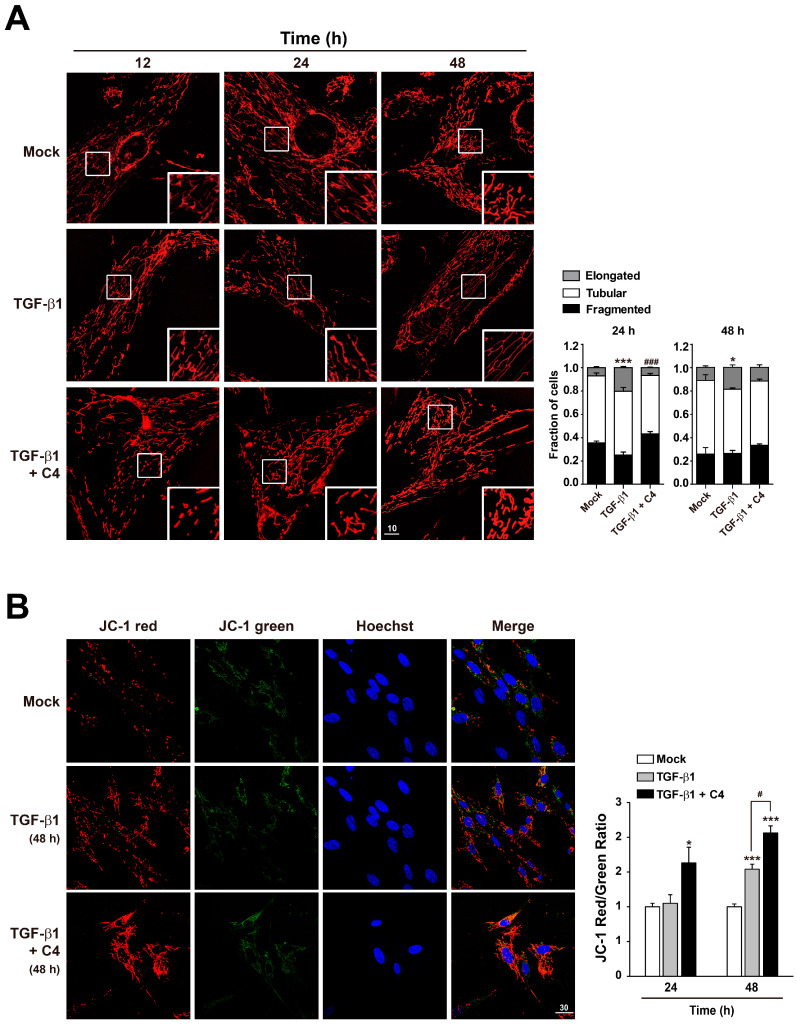
Effect of butyrate on mitochondrial morphology and function in TGF-β1-treated pulmonary fibroblasts. (**A**) MRC5 cells were stained with Hoechst 33,258 and Mitotracker Red after treatment with TGF-β1 (5 ng/mL) for the indicated times, with or without C4 (5 mM). Samples were analyzed by fluorescence microscopy. Scale bar denotes 10 μm. Representative images are shown. Data are means ± SEM; * *p* < 0.05 and *** *p* < 0.01; Mock vs. TGF-β1-treated cells, ^###^
*p* < 0.001; TGF-β1-treated cells vs. TGF-β1 + C4-treated cells. (**B**) MRC5 cells were stained with Hoechst 33,258 and JC-1 after treatment with TGF-β1 (5 ng/mL) for the indicated times, with or without C4 (5 mM). Samples were analyzed by fluorescence microscopy. Scale bar denotes 30 μm. Representative images are shown. Mean fluorescence intensities (MFI) of JC-1 staining were measured with FV10-ASW image analysis software (Olympus). Data are means ± SEM; * *p* < 0.05 and *** *p* < 0.001; Mock vs. TGF-β1-treated cells or TGF-β1 + C4-treated cells. ^#^
*p* < 0.05; TGF-β1-treated cells vs. TGF-β1 + C4-treated cells.

**Figure 5 metabolites-11-00258-f005:**
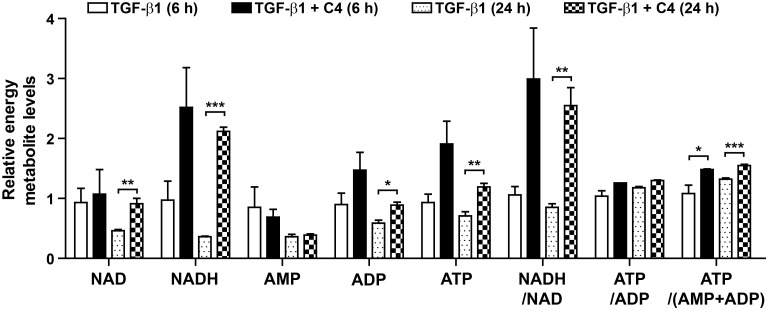
Effect of C4 on the levels of ATP, ADP, AMP, ATP/ADP, ATP/(ADP + AMP), NAD, NADH, and NADH/NAD in TGF-β1-treated pulmonary fibroblasts. MRC5 cells were treated with TGF-β1 (5 ng/mL) for the indicated times with or without C4 (5 mM). Data are expressed as means ± SEM of three independent experiments, * *p* < 0.05, ** *p* < 0.01, and *** *p* < 0.001; TGF-β1-treated cells vs. TGF-β1 + C4-treated cells.

## Data Availability

The data presented in this study are available in article and Appendix A.

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
