# Peer review of "Butyrate Prevents TGF-β1-Induced Alveolar Myofibroblast Differentiation and Modulates Energy Metabolism"

_metabolites, 2021, doi:10.3390/metabo11050258_

Round 1

Reviewer 1 Report

In this manuscript, the author reported the therapeutic effects from butyrate on TGF-beta1–induced alveolar myofibroblast differentiation. The mechanism was found to be mediated by cellular energy metabolism. These findings are interesting, however, I would like the author to improve the quality of this work by addressing the following concerns:

One of the key Figures, figure 1E, was not found in Figure 1. Please add it to the manuscript.

The concentrations of butyrate used in the cell studies are quiet high (1-10 mM). I am not sure about the endogenous content of butyrate in mammals. The author may want to provide some relevant discussion to address the concentration issues.

Butyrate was known to be an energy source in colonocytes. Please look into this aspect and provide more mechanistic insights.

Reviewer 2 Report

Lee et al. use a human lung fibroblast cell line to describe changes in energy metabolism during TGFβ-induced myofibroblast differentiation. The authors describe how TGFβ induced differentiation into myofibroblasts have a reduced oxidative phosphorylation among other changes in metabolism and subsequently show that the short chain fatty acid (SCFA) butyrate (C4) is able to prevent fibroblast differentiation into myofibroblasts as well as increase oxidative phosphorylation. Myofibroblast are important contributors to pulmonary fibrosis and idiopathic pulmonary fibrosis and are targets for therapeutic intervention.  The use of SCFAs to prevent differentiation is an interesting finding.

Overall, this is an interesting report that that highlights a potentially important finding describing the use of a SCFAs to prevent myofibroblast differentiation.  The work is logical, although there are a number of issues that the authors need to address to substantiate their claims.

Major comments:

  1. Figure 1E is missing from the manuscript. The data is the foundation to the authors’ study describing the baseline measurements energy metabolism including ATP, ADP, AMP, ATP/ADP, ATP (ADP+AMP), NAD, and NADH between the mock and TGFβ1 treated fibroblasts.

  1. In Figure 2B there are a few items for the authors to address. First, in the qPCR expression data for Col3A1 the TGFβ1+C4 (5mM) data is missing. Second, the data for mock and TGFβ1 conditions should be on the same graph and scale as the SCFA treatment groups to avoid misleading the reader. Third, error bars and statistics need to be added for the SCFA treatment groups.  If the conclusion is that a SCFA treatment reduces a gene transcript then TGFβ1 and TGFβ1+SCFA should be compared statistically.  To confirm a dose response each increasing concentration of SCFA should be statistically compared to the one prior.

  1. From the experiments with the SCFA, it would be important to know if the modulation of fibroblast energy metabolism or an alternative mechanism mediated by butyrate is altering TGFβ induced differentiation into myofibroblasts. This could be done in multiple way one of which is artificially altering the NADH/NAD+ ratio (oxidative phosphorylation) without butyrate under TGFβ1 treatment.

Minor comments:

  1. Title needs to describe the effect of butyrate. For example, “Modulation of energy metabolism by butyrate prevents TGFβ1-induced alveolar myofibroblast differentiation”

  1. On page 7 the description of “oxidative phosphorylation (Figure 1D)” is not correct and I assume will eventually be Figure 1E.

  1. Figure 5 should have a data from the mock treated group. This may not be important when the data from the mock treated group is present in Figure 1E.

Reviewer 3 Report

The authors aimed at investigating the effects of short chain fatty acids (SCAs) during TGFb1-induced pulmonary fibroblast differentiation into myofibroblasts with regard to fibrotic markers but also metabolites involved in energy metabolism. Among three different SCAs they find butyrate as the one suppressing the fibroblast differentiation induced by TGFb1 in parallel with increased mitochondrial membrane potential and elevated levels of metabolites.

The rational why SCAs in lung should be explored is not very clear. The explanation for why SCAs are present in the lung is sparsely described. Saying “SCFAs are likely to be present in lung tissues” indicates that it is not yet proven that SCAs even exist in lung tissues. My question here is why the authors think it is important to study SCAs in lung fibrosis. I think this should be better explained more emphazized.

Other comments that I have. The descriptions in the main text for mRNA levels of Col1A1, Col3A1, α-SMA, and Fibronectin and immunofluorescence analysis of Col1A1 and α-SMA (Figure 1C,D) are very brief and observations should be explained and discussed in more detail.

Figure 1 contains only A, B and C. However, the legend of Figure 1 contains a description for E which is not shown. This has to be corrected. This is the same for the main text. Figure 1 E is also mentioned/described but the figure itself is missing. Thus, it made it difficult to follow as subsequent experiments were based on those results.

Round 2

Reviewer 1 Report

In general, the authors have addressed my concerns from the previous round of reviewing. As such, I support the publication of this revised manuscript.

Author Response

We really appreciate your support.

Reviewer 2 Report

The authors have addressed all my comments and made significant advancement in the manuscript. 

Only very minor comment is in the figure legend 2.  The description of the statistics is provided twice.  The second description in purple text in my copy at the end of the legend is the most clear.
